# Obtaining of Mg-Zn Co-Doped GaN Powders via Nitridation of the Ga-Mg-Zn Metallic Solution and Their Structural and Optical Properties

**DOI:** 10.3390/ma16083272

**Published:** 2023-04-21

**Authors:** Erick Gastellóu, Rafael García, Ana M. Herrera, Antonio Ramos, Godofredo García, Gustavo A. Hirata, José A. Luna, Roberto C. Carrillo, Jorge A. Rodríguez, Mario Robles, Yani D. Ramírez, Guillermo Martínez

**Affiliations:** 1División de Sistemas Automotrices, Universidad Tecnológica de Puebla (UTP), Antiguo Camino a la Resurrección 1002–A, Zona Industrial, Puebla 72300, Puebla, Mexico; 2Departamento de Investigación en Física, Universidad de Sonora (UNISON), Rosales y Colosio, C. De la Sabiduría, Centro, Hermosillo 83000, Sonora, Mexico; 3Centro de Investigación en Dispositivos Semiconductores, Benemérita Universidad Autónoma de Puebla (BUAP), 14 Sur y Av. San Claudio, Puebla 72570, Puebla, Mexico; 4Centro de Nanociencias y Nanotecnología, Universidad Nacional Autónoma de México (UNAM), Carr. Tijuana-Ensenada km107, C.I.C.E.S.E., Ensenada 22860, Baja California, Mexico; 5Departamento de Física, Universidad de Sonora (UNISON), Rosales y Colosio, C. De la Sabiduría, Centro, Hermosillo 83000, Sonora, Mexico; 6Departamento de Investigación y Desarrollo, Universidad Tecnológica de Puebla (UTP), Antiguo Camino a La Resurrección 1002–A, Zona Industrial, Puebla 72300, Pueble, Mexico; 7Departamento de Ingeniería Química, Universidad de Guanajuato (UGTO), Noría Alta S/N, Guanajuato 36050, Guanajuato, Mexico

**Keywords:** co-doped, GaN, nitridation, liquid metallic solution, semiconductors

## Abstract

Mg-Zn co-dopedGaN powders via the nitridation of a Ga-Mg-Zn metallic solution at 1000 °C for 2 h in ammonia flow were obtained. XRD patterns for the Mg-Zn co-dopedGaN powders showed a crystal size average of 46.88 nm. Scanning electron microscopy micrographs had an irregular shape, with a ribbon-like structure and a length of 8.63 µm. Energy-dispersive spectroscopy showed the incorporation of Zn (Lα 1.012 eV) and Mg (Kα 1.253 eV), while XPS measurements showed the elemental contributions of magnesium and zinc as co-dopant elements quantified in 49.31 eV and 1019.49 eV, respectively. The photoluminescence spectrum showed a fundamental emission located at 3.40 eV(364.70 nm), which was related to band-to-band transition, besides a second emission found in a range from 2.80 eV to 2.90 eV (442.85–427.58 nm), which was related to a characteristic of Mg-doped GaN and Zn-doped GaN powders. Furthermore, Raman scattering demonstrated a shoulder at 648.05 cm^−1^, which could indicate the incorporation of the Mg and Zn co-dopants atoms into the GaN structure. It is expected that one of the main applications of Mg-Zn co-doped GaN powders is in obtaining thin films for SARS-CoV-2 biosensors.

## 1. Introduction

The semiconductor supply chain (SSC) is very important nowadays and is vital in the industry’s demand, such as computers, communication, consumer, automotive, and government. The SARS-CoV-2 pandemic had devastating effects on SSC, generating a shortage in the semiconductor industry, besides increased prices in electronic devices [1]. Currently, the semiconductor industry is beginning to activate, where some of the materials more affected have been the III-nitride compounds. Gallium nitride (GaN) is one of the more important binary III-nitride compounds, with structural and optical properties, whose characteristics are continuing to improve through research. The doping of the GaN is attractive owing to changes in the mobility of charge carriers. GaN has applications in full-color displays, GaN-based power devices, Schottky barrier diodes, light-emitting diodes, high-electron-mobility transistors, piezoelectric MEMs, and solar cells [2,3,4]. Other applications could be in laser devices for optical communications; medicine; and industry [5,6,7,8,9].

In recent years, the strategies of doping and of defect control have regained their importance in semiconductor physics for researchers. Co-doping is an important strategy used for tuning the dopant in electronic and magnetic properties, besides enhancing dopant solubility, and increasing activation by lowering the ionization energy of acceptors and donor, thus increasing the carrier’s mobility [10,11]. In P-type GaN, the Mg-H co-doped is one of the best examples of fully compensated co-doping [11]. K.H. Ploog investigated the Be-O co-dopedGaN with a strongly improved P-type conductivity at room temperature due to the substantial enhancement of the hole-mobility [12]. On the other hand, R.Y. Korotkov et al. investigated Mg-O co-dopedGaN, obtaining hole concentrations de 2 × 10^18^ cm^−3^ at 295 K, with resistivity from 8 to 0.2 Ωcm [13]. In another work, H. Pan et al. showed a novel application of photocatalytic activity in the visible light region realized with Cr-O co-dopedGaN [14]. P-type GaN can be obtained by incorporating divalent elements such as Zn, Be, and Mg, where the co-doping of two different dopants could produce a low resistivity P-type GaN. K.S. Kim et al. showed the obtaining of Mg-Zn co-dopedGaN grown by MOCVD, showing a P-type GaN with low electrical resistivity of 0.72 Ωcm and a high hole concentration of 8.5 × 10^17^ cm^−3^ [15,16]. In general, there is poor research in the literature regarding the Mg-Zn co-dopedGaN material.

This investigation shows the synthesis of Mg-Zn co-dopedGaN powders via the nitridation of the Ga-Mg-Zn metallic solution in ammonia flow at 1000 °C for 2 h, which could have application in the deposit of thin films by RF magnetron sputtering using laboratory-prepared targets. This is a way of contributing to research on the co-doping of GaN with Mg-Zn. To compare the results obtained of Mg-Zn co-dopedGaN powders, undoped GaN powders were obtained by the nitridation of metallic gallium, as was reported in our before studies [17]. On the other hand, the characterizations X-ray diffraction patterns (XRD), scanning electron microscopy (SEM), energy-dispersive spectroscopy (EDS), X-ray photoelectron spectroscopy (XPS), transmission electron microscopy (TEM), photoluminescence (PL), and Raman scattering were carried out to know the structural and optical properties of the Mg-Zn co-dopedGaN powders.

## 2. Materials and Methods

Mg-Zn co-dopedGaN powders were synthesized using metallic gallium (99.999%), metallic magnesium, and metallic Zn as reagents. Moreover, as the source of nitrogen atoms, ammonia (NH_3_) was used. To obtain the Mg-Zn co-dopedGaN, the powders were used 1.32 g of metallic gallium (19.02 mmol), 5.20 mg of metallic magnesium (0.21 mmol ≅ 0.4%), and 7.80 mg of metallic zinc (0.12 mmol ≅ 0.6%). To obtain the undoped GaN powders, 3.37 g (48.41 mmol) of metallic gallium was used, following the process realized in our before work [18]. The reaction formulated to obtain the Mg-Zn co-dopedGaN material is the following [17]:(1)$$2\mathrm{Ga}-\mathrm{Mg}\left(l\right)-\mathrm{Zn}\left(l\right)+2{\mathrm{NH}}_{3}\left(g\right)\to 2\mathrm{GaN}:\mathrm{MgZn}\left(s\right)+3H_{2}\left(g\right)$$

### 2.1. Mg-Zn Co-Doped GaN Powders

To begin the process of synthesis of Mg-Zn co-dopedGaN powders, the general preparation of the metallic solution was carried out according to the work of Gastellou et al. [18]. Once obtained, the metallic solution was introduced inside a chemical vapor deposition furnace (CVD), whereupon the CVD system was purged, and later an N_2_ flow at 50 sccm was opened and was allowed to flow into the atmosphere. After, to ensure the diffusion of Zn atoms into the gallium of the Ga-Zn liquid solution, the temperature was increased to 440 °C for 40 min [18,19]. On the other hand, to ensure the diffusion of Mg atoms into the metallic gallium of the Ga-Mg-Zn metallic solution, the temperature was increased to 670 °C for 40 min (20 °C above the Mg melting point) [20,21]. Later, once the temperature had been stabilized at 670 °C and 40 min has passed, the N_2_ flow was closed, and an NH_3_ flow at 150 sccm was introduced inside the furnace. At this moment, the temperature of the Ga-Mg-Zn metallic solution was increased to 900 °C for 40 min for its homogenization process (first stage), and finally the Ga-Mg-Zn metallic solution was increased to 1000 °C for 2 h for its nitridation process (second stage). After the nitridation process was finished, the ammonia flow was closed; then, the temperature was decreased to room temperature using an N_2_ flow at 150 sccm to cool the system. The synthesized material was taken out of the furnace and ground to obtain M-Zn co-dopedGaN powders.

The synthesis does not require a cleaning process due to the fact the Mg atoms and Zn atoms were defunded in the Ga metallic and transformed into the final material. The total weight synthesized of Mg-Zn co-dopedGaN powders was 1.5661 g (9.02 mmol), which indicates that there was a nitrogen incorporation of 233.10 mg (16.65 mmol). A diagram of the synthesis process is shown in Figure 1, while Table 1 shows the parameters considered in the synthesis of Mg-Zn co-dopedGaN powders. 

### 2.2. Characterizations

The Mg-Zn co-dopedGaN and undoped GaN powders were characterized by X-ray diffraction (XRD) using a Philips X’PERT MPD equipment with a wavelength (Cu Kα) of 1.5406 Å. The measurements of XRD patterns were realized with a range from 30° to 60°. The surface morphology and the elemental analysis (SEM/EDS) of the Mg-Zn co-dopedGaN, and of the undoped GaN powders, were obtained using JEOL JSM-7800F Schottky Field Emission equipment. X-ray photo-electron spectroscopy (XPS) characterizations were carried out using an Escalab 250Xi Brochure equipment with an energy range from 0 to 1400 eV for the Mg-Zn co-dopedGaN powders. Transmission electron microscopy (TEM) was obtained using JEOL JEM-2010 equipment for the Mg-Zn co-dopedGaN and the undoped GaN powders. Photoluminescence spectra (PL) were measured at room temperature with an excitation wavelength of 325 nm (UV) and a power of 55 mW, using an IK series He–Cd LASER for the Mg-Zn co-dopedGaN, and undoped GaN powders. Finally, the Raman scattering characterizations for the Mg-Zn co-dopedGaN and the undoped GaN powders were obtained using a Horiba Jobin Yvon HR-800 Micro Raman spectrophotometer.

## 3. Results and Discussion

### 3.1. Structure

Figure 2 shows a comparison between the XRD patterns of the undoped GaN powders (Figure 2a), and the Mg-Zn co-dopedGaN powders (Figure 2b). In Figure 2, all of the peaks of both X-ray diffraction patterns were indexed in the ICDD pdf card: 00-050-0792. The **a** peak was found at plane orientation (100), **b** at (002), **c** at (101), **d** at (102), and **e** at (110). The lattice constants a = 3.1890 Å and c = 5.1855 Å were calculated for the hexagonal structure, with a ratio c/a of 1.626 belonging to space group P6_3_mc(186). Nitrides oxide or pure metals were not detected, indicating the adequate diffusion of the co-dopants of Mg-Zn into GaN powders. Moreover, Figure 2 shows that there is no a significant difference between the diffraction patterns Figure 2a,b. Both diffraction patterns showed narrowed peaks, which could indicate the presence of large crystals in both samples. FWHM measurements were carried out for the X-ray diffraction pattern of Figure 2b; the (100) plane orientation had 0.1634°, 0.1708° for (002), 0.1786° for (101), 0.2130° for (102), and 0.2424° for (110). Table 2 shows the measurements for peak position, FWHM, crystal size, and interplanar spacing for Figure 2a,b. Calculations of the displacement of the Figure 2b in relation to Figure 2a based on Table 2 were carried out, finding a displacement to the right. The **a** peak had a displacement of 0.0493°, the **b** peak of 0.0527°, the **c** peak of 0.0513°, the **d** peak of 0.0527°, and the **e** peak of 0.0486°. Using the ICCD PDF-4+ 2022 software and the Debye-Scherrer [22], the crystal size had a value of 48.61 nm for the undoped GaN powders and 46.88 nm for the Mg-Zn co-dopedGaN powders.

### 3.2. Electron Microscopy

Figure 3a shows the surface morphology of the undoped GaN, which demonstrated hexagonal crystals for the micrograph at 2500×, whose values were 20.76 μm in width and 23.16 μm in length. Moreover, the micrograph at 5000× also showed hexagonal crystals of different sizes with values of 6.28 μm in width and 7.80 in length. On the other hand, Figure 3b showed the surface morphology of the Mg-Zn co-dopedGaN powders, which had an irregular shape with a ribbon-like structure with an average length of 8.63 μm. The surface morphology of the ribbon-like structure of the Figure 3b could be related to the Mg and Zn incorporation via diffusion into the GaN. It is possible that during the formation of the ribbon-like structure, oxygen or carbon interstitial atoms are desorbed of the GaN structure, whereupon magnesium or zinc replace the vacancies; this way, the co-doping with Mg and Zn into the GaN structure is carried out.

Figure 4a presents the energy dispersive spectroscopy (EDS) spectrum corresponding to Figure 3a, which demonstrated the elemental contributions of gallium and nitrogen; besides copper belonging to the sample holder, there are few traces of non-intentional impurities of oxygen and carbon. Figure 4b shows the EDS spectrum corresponding to Figure 3b, which showed the elemental contributions of gallium (Kα 9.241 eV and Lα 1.098 eV), and nitrogen (Kα 0.392 eV). Figure 3b also showed few traces of oxygen and carbon. It is important to mention that Figure 3b demonstrated the presence of zinc (Lα 1.012 eV) and magnesium (Kα 1.253 eV), with atomic percentages of 0.60% and 0.32%, respectively. Furthermore, the atomic percentages for gallium and nitrogen of Figure 3b were 49.66% and 31.81%, respectively.

Figure 5a shows the TEM micrograph of the sample of Mg-Zn co-dopedGaN powders, which had values of 496.5 nm in width and 606.8 nm in length, for a magnification at 200 nm. Figure 5b shows the electron diffraction pattern for the undoped GaN powders, which can be observed as a distribution uniform of GaN atoms. However, Figure 5c showed the electron diffraction pattern for the Mg-Zn co-dopedGaN powders, where a greater scattering of impurities or co-dopant atoms in the GaN sample was observed. This scattering of impurities might indicate the incorporation of the diffusion of Mg and Zn atoms into GaN, which agrees with the ribbon-like structure in Figure 3b and the EDS spectrum in Figure 4b.

### 3.3. X-ray Photo-Electron Spectroscopy

Figure 6 presents the XPS spectra of the Mg-Zn co-doped GaN powders. Figure 6a presents the peaks for high energies of Ga 2P_1/2_ and Ga 2P_3/2_, with values of 1146.28 eV and 1119.76 eV, respectively. Figure 6b depicts the N 1s peak with an energy value of 399.29 eV. Figure 6c shows the Zn 2P_3/2_ peak with an energy value of 1019.49 eV, whose binding energy belongs to L_3_ level. Figure 6d depicts the Mg 2P_3/2_ peak with a binding energy of 49.31 eV. It is important to mention that during the obtaining process of the Mg-Zn co-doped GaN powders, it was attempted to reduce the presence of oxygen and carbon, which can act as non-intentional impurities. However, Figure 6e shows the elemental contribution of the O 1s peak with a binding energy of 532.72 eV, and the Figure 6f shows the presence of the elemental contribution of C 1s with a value of 285.68 eV, which could affect the optical properties of Mg-Zn co-doped GaN.

### 3.4. Photoluminescence

Figure 7 shows the PL spectrum of the undoped GaN powders (black curve) and the PL spectrum of the Mg-Zn co-doped GaN powders (red curve). The **a** peak depicts high emission located at 3.40 eV (364.70 nm) due to the band-to-band transition, which corresponds to the ultraviolet emission for the hexagonal GaN. The **b** peak had an emission band located at 3.34 eV (371.25 nm) corresponding to the undoped GaN powders, which had been observed in samples with excitons bound to structural defects of GaN, perhaps related to the high concentration of stacking faults [23]. The ***c*** peak corresponds to the blue emission band located in a range from 2.80 eV to 2.90 eV (442.85–427.58 nm), which is characteristic of magnesium-doped GaN and zinc-doped GaN [24,25]. Figure 8 shows the decomposition of the **c** peak, where is possible to observe the emission of 2.90 eV (**g** peak) related to magnesium-doped GaN. Furthermore, Figure 8 also shows the emission of 2.80 eV (**h** peak) related to zinc-doped GaN. The **d** peak was located in a yellow emission band with an energy of 2.22 eV (558.14 nm) for the undoped GaN powders, which was related to Ga vacancies (V_Ga_) and the substitutional atoms of carbon or oxygen (Figure 3a), indicating the obtaining of N-type GaN. On the other hand, the yellow emission band for the Mg-Zn co-dopedGaN powders was null due to the P-type GaN samples [25]. The ***e*** peak had an emission with energy located at 1.70 eV (729.41 nm). This red emission band has been observed in heavily Mg-doped P-type GaN [23]. Finally, the **f** peak showed an emission in the red part of the spectrum at 1.67 (742.51 nm), which has been observed in Ga-lean samples. Figure 9a shows the yellow luminescence in undoped GaN powders, while Figure 9b shows the violet-blue luminescence of the Mg-Zn co-doped GaN powders, which were obtained with the excitation of a UV lamp “Blak-Ray UVL-56 (366 nm)”.

### 3.5. Raman Scattering

Figure 10a shows the Raman spectrum for the undoped GaN powders (red curve), and Figure 10b shows the Raman spectrum for the Mg-Zn co-doped GaN powders (black curve). Both Raman spectra show the classical vibration modes A_1_(TO) and E_2_(high) for the hexagonal GaN structure. However, the Raman spectrum for the Mg-Zn co-dopedGaN powders also showed the vibration mode E_2_(low). The E_2_(high) vibration mode has a similar frequency for the undoped GaN and Mg-Zn co-dopedGaN powders, with a value of 557.38 cm^−1^. The vibration mode A_1_(TO) for undoped GaN powders had a frequency of 525.34 cm^−1^, while the vibration mode A_1_(TO) for Mg-Zn co-dopedGaN powders was 521.15 cm^−1^, which depicts a slight shift to the left of 4.19 cm^−1^ for the Mg-Zn co-dopedGaN powders compared to undoped GaN powders. This slight shift of the phononic vibration mode could be related to the incorporation by the diffusion of magnesium and zinc co-dopants atoms into GaN [24,25]. Furthermore, a shoulder was located at 648.05 cm^−1^ (Figure 10b), which also might be indicative of the incorporation of magnesium and zinc co-dopants atoms into GaN powders.

## 4. Conclusions

Mg-Zn co-doped GaN powders were obtained via nitridation of a metallic solution of Ga-Mg-Zn at 1000 °C for two hours in ammonia flow. XRD diffraction patterns for Mg-Zn co-doped Ga powders showed a crystal size average of 46.88 nm, with a shift towards greater angles compared to undoped GaN powders. The SEM micrographs demonstrated an irregular shape with a ribbon-like structure and an average length of 8.63 µm for Mg-Zn co-doped GaN powders. EDS showed the presence of zinc (Lα 1.012 eV) and magnesium (Kα 1.253 eV) as co-dopants atoms, while XPS characterization showed the presence of elemental contributions of magnesium (49.31 eV) and zinc (1019.49 eV), which agree with the EDS analysis. TEM micrographs showed a greater scattering of impurity atoms for the Mg-Zn co-doped GaN powders compared to undoped GaN powders. The photoluminescence spectrum for Mg-Zn co-doped GaN powders showed two fundamental emissions: the first emission located at 3.40 eV(364.70 nm) related to band-to-band transition for hexagonal GaN and a second emission located in a range from 2.80 eV to 2.90 eV (442.85–427.58 nm), which are values characteristics of Mg-doped GaN and Zn-doped GaN powders. Raman scattering demonstrated a shoulder at 648.05 cm^−1^, which could indicate the incorporation of the Mg and Zn co-dopants atoms into the GaN structure.

## Figures and Tables

**Figure 1 materials-16-03272-f001:**
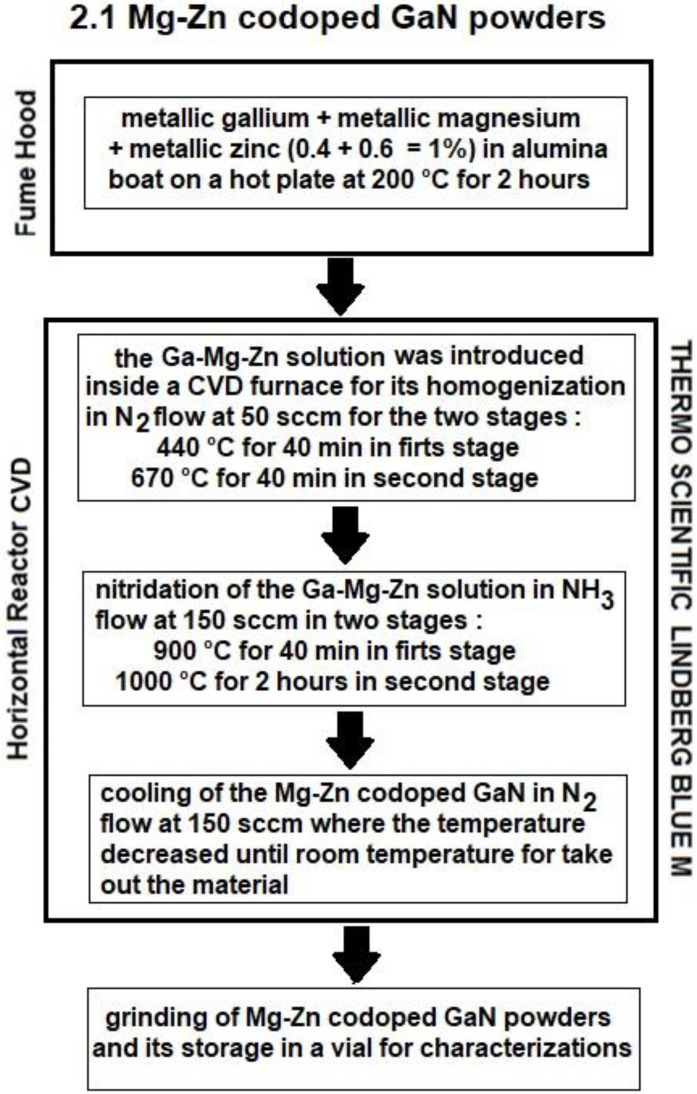
Diagram to obtain the Mg-Zn co-dopedGaN powders.

**Figure 2 materials-16-03272-f002:**
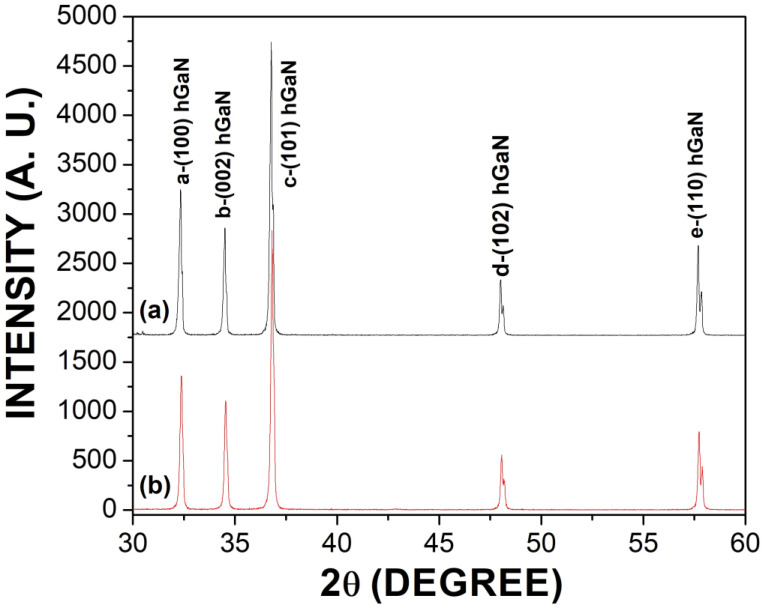
(**a**) XRD pattern of the undoped GaN powders; (**b**) XRD pattern of the Mg-Zn co-doped GaN powders.

**Figure 3 materials-16-03272-f003:**
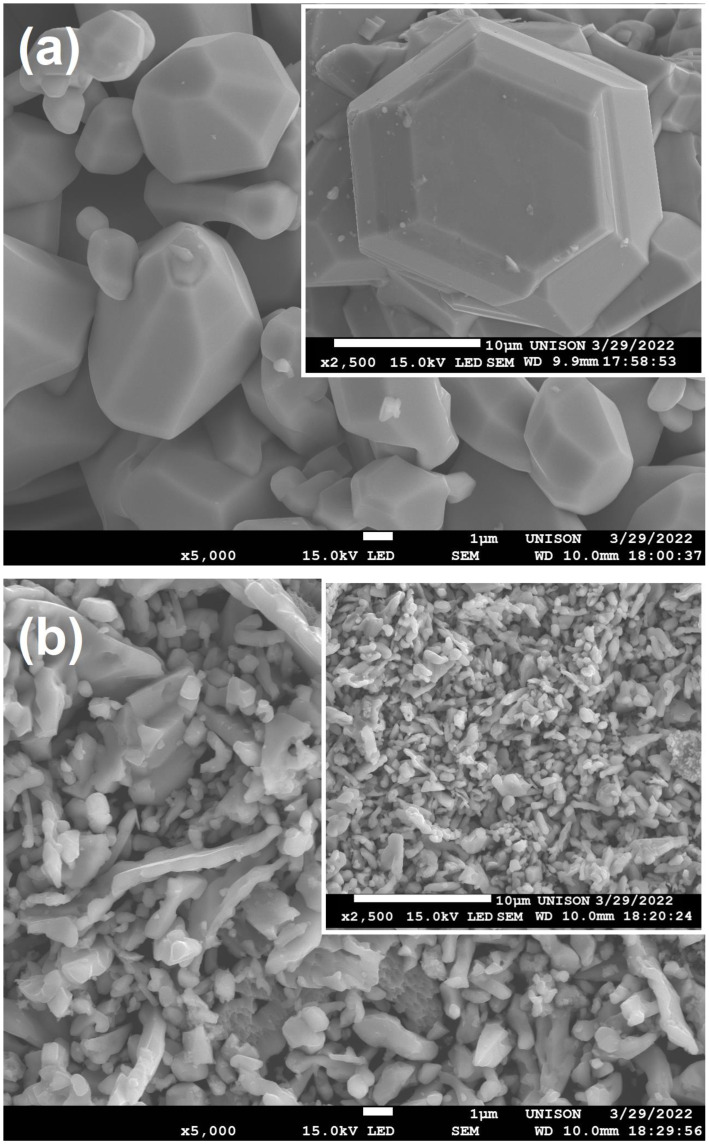
(**a**) SEM micrographs of the undoped GaN powders; (**b**) SEM micrographs of the Mg-Zn co-doped GaN powders.

**Figure 4 materials-16-03272-f004:**
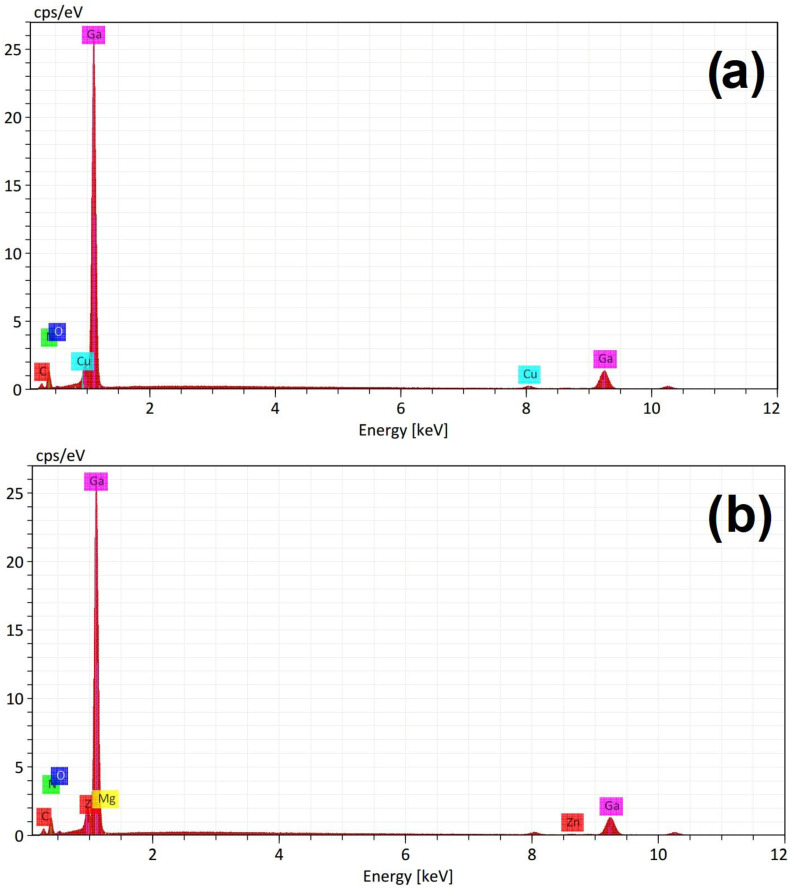
(**a**) EDS spectrum of SEM image of Figure 3a for the undoped GaN powders, (**b**) EDS spectrum of SEM image of Figure 3b for the Mg-Zn co-doped GaN powders.

**Figure 5 materials-16-03272-f005:**
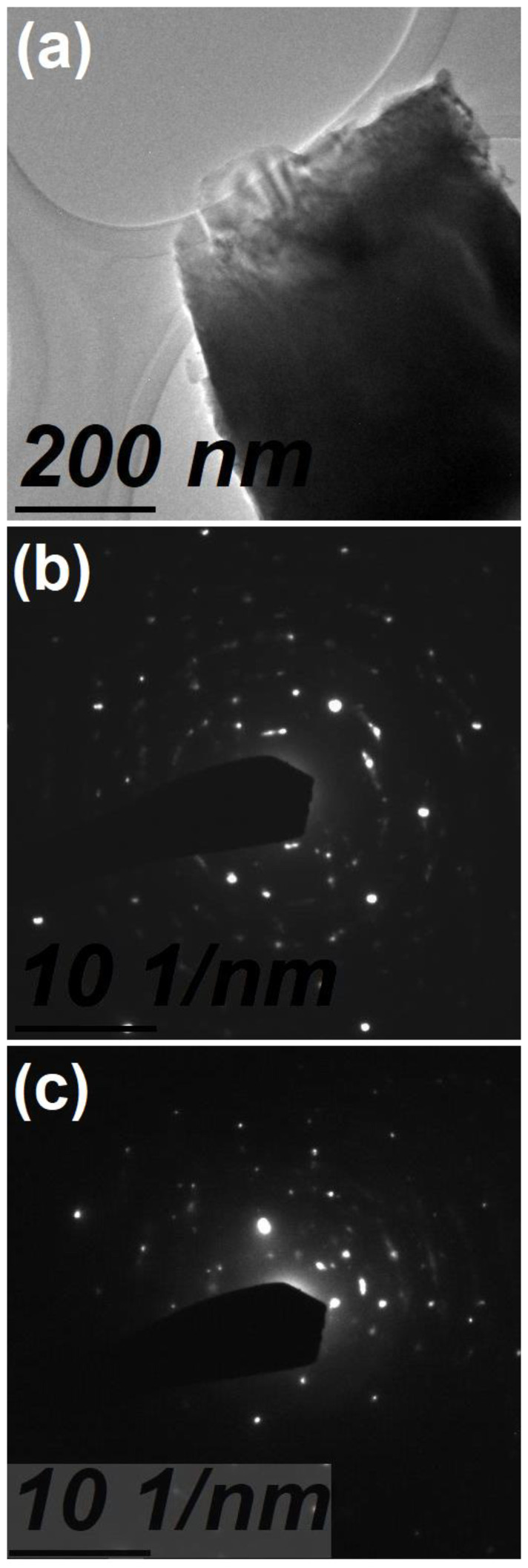
(**a**) TEM micrograph of the sample of Mg-Zn co-doped GaN powders, (**b**) electron diffraction pattern of the undoped GaN powders, and (**c**) electron diffraction pattern of the Mg-Zn co-doped GaN powders.

**Figure 6 materials-16-03272-f006:**
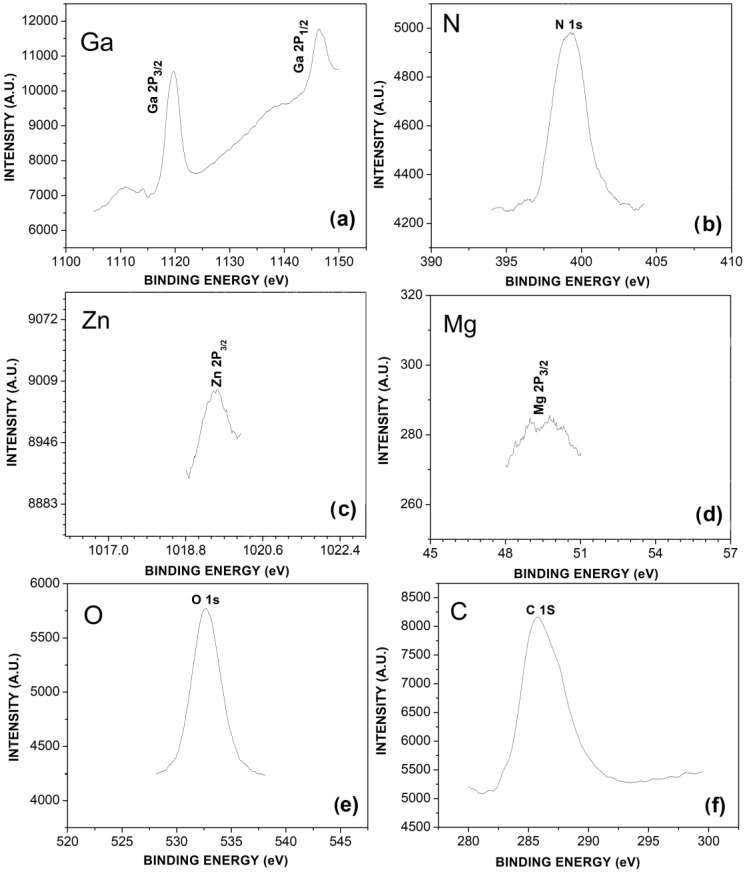
XPS spectra of the Mg-Zn co-doped GaN powders:(**a**) Ga 2P_3/2_ and Ga 2P_1/2_, (**b**) N 1s, (**c**) Zn 2P_3/2_, (**d**) Mg 2P_3/2_, (**e**) O 1s, and (**f**) C 1s peaks.

**Figure 7 materials-16-03272-f007:**
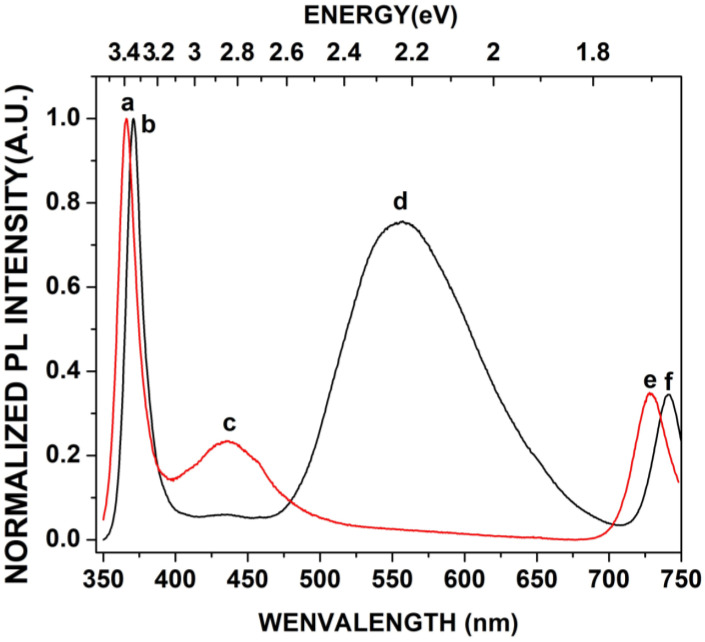
Photoluminescence spectra of the undoped GaN powders and the Mg-Zn co-doped GaN powders (a–f).

**Figure 8 materials-16-03272-f008:**
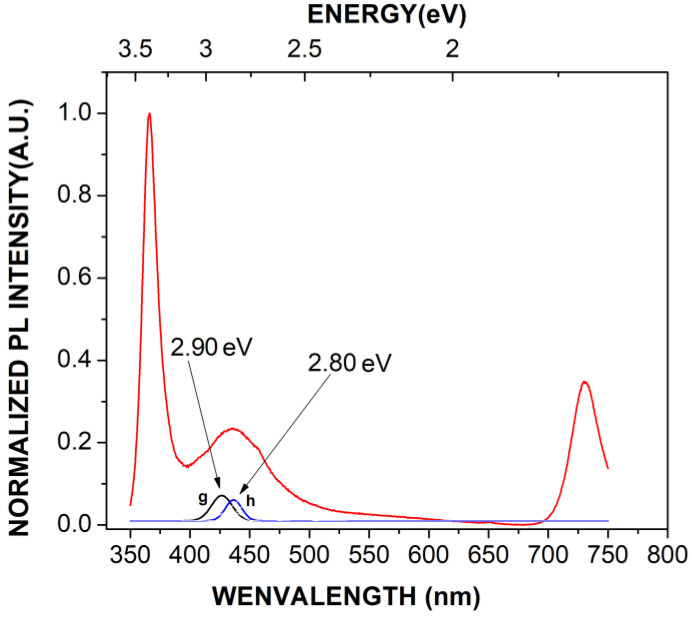
Decomposition of the **c** peak of the photoluminescence spectrum of the Mg-Zn co-doped GaN powders of Figure 7.

**Figure 9 materials-16-03272-f009:**
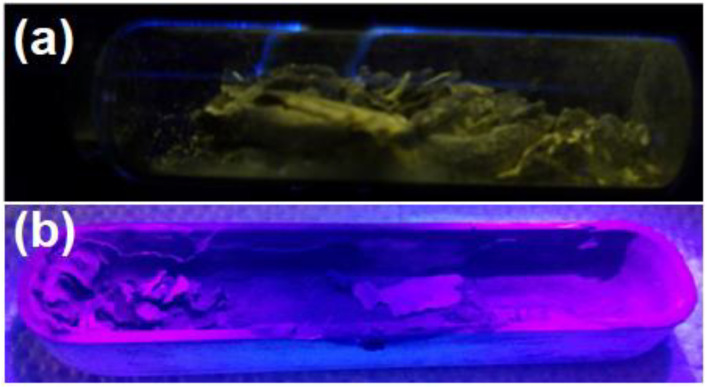
(**a**) Yellow luminescence of the undoped GaN powders; (**b**) violet-blue luminescence of the Mg-Zn co-dopedGaN powders.

**Figure 10 materials-16-03272-f010:**
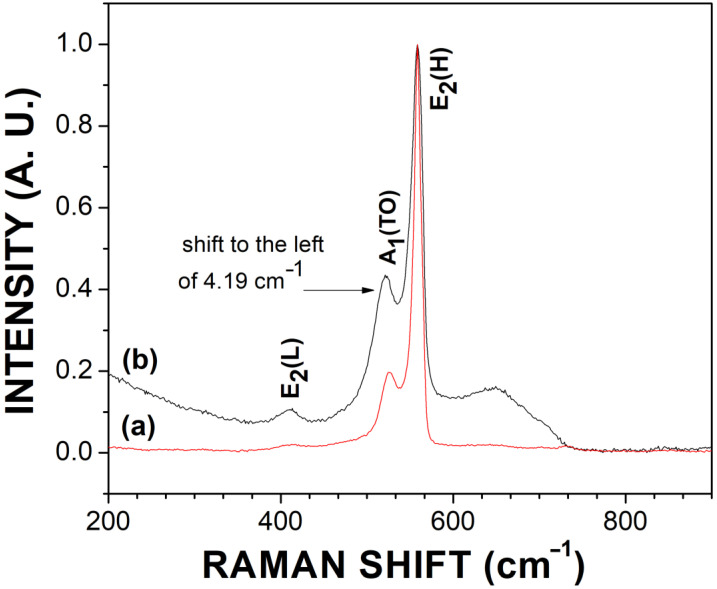
(**a**) Raman scattering of the undoped GaN powders; (**b**) Raman scattering of the Mg-Zn co-doped GaN powders.

**Table 1 materials-16-03272-t001:** Parameters considered in the synthesis of Mg-Zn co-dopedGaN powders.

Ga Weight (g)	Mg Weight (mg)	Zn Weight (mg)	Zn Homogenization Temperature (°C)	Mg Homogenization Temperature (°C)	Nitridation Temperature (°C)	Nitridation Time (h)
1.32	5.20	7.80	440	670	1000	2

**Table 2 materials-16-03272-t002:** Values calculated for peak position, FWHM, crystal size, and interplanar spacing for undoped GaN and Mg-Zn co-dopedGaN powders.

	Undoped GaN Powders				Mg-Zn Co-Doped GaNPowders			
Peak	Peak Position (Degree)	FWHM (Degree)	Crystal Size (nm)	Interplanar Spacing (Å)	Peak Position (Degree)	FWHM (Degree)	Crystal Size (nm)	Interplanar Spacing (Å)
**a**	32.3247	0.1697	50.8913	2.7672	32.3740	0.1634	52.8691	2.7631
**b**	34.4971	0.1615	53.7662	2.5978	34.5498	0.1708	50.8475	2.5939
**c**	36.7747	0.1742	50.1737	2.4419	36.8260	0.1786	48.9618	2.4386
**d**	48.0318	0.1956	46.4284	1.8926	48.0845	0.2130	42.6433	1.8907
**e**	57.7078	0.2266	41.7941	1.5962	57.7565	0.2424	39.0862	1.5949

## Data Availability

Data are contained within the article.

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
