# Peer review of "Obtaining of Mg-Zn Co-Doped GaN Powders via Nitridation of the Ga-Mg-Zn Metallic Solution and Their Structural and Optical Properties"

_materials, 2023, doi:10.3390/ma16083272_

Round 1
Reviewer 1 Report
Comment to the Author
The presented work entitled "Obtaining of Mg-Zn codoped GaN powders via nitridation of 2 the Ga-Mg-Zn metallic solution and their structural, and opti-3 cal properties" is based on experimental results which show the Mg-Zn codoped GaN powders via nitridation of a Ga-Mg-Zn metallic solution at 1000 °C 27 for two hours in NH3 flow were obtained. X-ray diffraction patterns for the Mg-Zn codoped GaN 28 powders showed a crystal size average of 46.88 nm. Scanning electron microscopy micrographs 29 showed an irregular shape with a structure like ribbons and an average length of 8.63 μm. The manuscript may accept after incorporation of the following concerns, but the current form is not actable.
1. Abstract should rewrite as it should be indicated the main findings (with some values).
2. Abstract should contain the future application of work.
3. Introduction part should have more recent references.
4. At the last paragraph of introduction part, work should summarize.
5. Equations are not cited properly. Give the refence for each equation.
6. Process diagram should contain all steps for obtaining the Mg-Zn codoped GaN powders.
7. What parameters have been considered in this work. Show in a table.
8. In some of the sentence, it is difficult to understand the meaning of text.
9. Result explanation is not sufficient and justified. Give proper reasons for all the results and justify.
10. Conclusion should have some resulting details with numerical data.
Author Response
March 28th, 2023
Reviewer 1
Manuscript ID: materials-2270136
Dear Reviewer 1
Enclosed you will find the revised version of the manuscript: “Obtaining of Mg-Zn codoped GaN powders via nitridation of the Ga-Mg-Zn metallic solution and their structural, and optical properties", by Erick Gastellóu, Rafael García, Ana M. Herrera, Antonio Ramos, Godofredo García, Gustavo A. Hirata, José A. Luna, Roberto C. Carrillo, Jorge A. Rodríguez, Mario Robles, Yani D. Ramírez and Guillermo Martínez. We have implemented several changes in the text to answer all indications requested by the assistant editor and reviewer. We believe that their observations allowed us to improve substantially the quality of our work, and hence we manifest our gratitude to the professional job they did in the review of our manuscript.
In the revised manuscript, the yellow underlined sentences correspond to the corrections of the first reviewer; in addition to this, a thorough review of the manuscript was carried out taking into account the comments of the first reviewer. The manuscript was also proofread by a native grammar checker.
Yours Sincerely
Dr. Erick Gastellóu
erick_gastellou@utpuebla.edu.mx
Corresponding author
Comments of the first reviewer:
- Abstract should rewrite as it should be indicated the main findings (with some values).
Answer: The abstract was rewritten based on the comment of reviewer number 1.
- Abstract should contain the future application of work.
Answer: An application of the material was included in the abstract based on comment from reviewer number 1.
- Introduction part should have more recent references.
Answer: References in introduction 5 to 10 were included as most recent references based on reviewer number 1's recommendation.
- At the last paragraph of introduction part, work should summarize.
Answer: The last paragraph of the introduction was simplified according to reviewer number 1's comment.
- Equations are not cited properly. Give the refence for each equation.
Answer: The paragraph of equation (1) was rewritten according to the comment of reviewer number 1.
- Process diagram should contain all steps for obtaining the Mg-Zn codoped GaN powders.
Answer: Figure 1 was modified based on comment from reviewer number 1.
- What parameters have been considered in this work. Show in a table.
Answer: A table with the parameters considered for the process of obtaining Mg-Zn doped GaN powders was incorporated into the manuscript, according to the comment of reviewer number 1.
- In some of the sentence, it is difficult to understand the meaning of text.
Answer: The entire manuscript was revised with the aim of improving the writing.
- Result explanation is not sufficient and justified. Give proper reasons for all the results and justify.
Answer: The description of all results throughout the manuscript was justified as indicated by reviewer number 1.
- Conclusion should have some resulting details with numerical data.
Answer: The conclusion was modified according to reviewer number 1's comment, this adding more important data from the characterizations.

Reviewer 2 Report
Title: Obtaining of Mg-Zn codoped GaN powders via nitridation of the Ga-Mg-Zn metallic solution and their structural and optical properties
Authors: Erick Gastellóu, Rafael García, Ana M. Herrera, Antonio Ramos, Godofredo García, Gustavo A. Hirata, José A. Luna, Roberto C. Carrillo, Jorge A. Rodríguez, Mario Robles, Yani D. Ramírez and Guillermo Martínez
The manuscript by E. Gastellóu et al. presents the synthesis of Mg-Zn codoped GaN powders via nitridation of the Ga-Mg-Zn metallic solution. The results of this study are interesting from the point of view of practical use, in particular, for developing violet-blue emitters. The paper is very clearly written. In my view, it can be recommended for publication.
The only point should be addressed:
Chapter 3.4, lines 231-250; Conclusions, lines 285-289:
The authors claim that “the photoluminescence spectrum for Mg-Zn co-doped GaN powders showed two fundamental emissions, the first located at 3.40 eV (364.70 nm) related to band-to-band transition, and a second emission located in a range from 2.80 eV to 2.88 eV (442.85 nm – 430.55 nm), which is characteristic of Mg-doped GaN and Zn-doped GaN powders”.
It is known that Mg introduces two acceptor levels in GaN. The acceptor bound excitons give two PL peaks at 3.466 eV (unstable in p-GaN) and 3.454 eV. The most common radiation bands in Mg-doped GaN are the BL bands with the peak of about 2.9 eV, and the UVL band, whose peak is about 3.2 eV. Sometimes, the GL band of about 2.4-2.5 eV can appear. It seems that the authors do not identify all the bands related with Mg impurity in GaN. Moreover, it is not entirely clear whether the authors distinguish the difference between the bands associated with Mg and Zn.
Author Response
March 28th, 2023
Reviewer 2
Manuscript ID: materials-2270136
Dear Reviewer 2
Enclosed you will find the revised version of the manuscript: “Obtaining of Mg-Zn codoped GaN powders via nitridation of the Ga-Mg-Zn metallic solution and their structural, and optical properties", by Erick Gastellóu, Rafael García, Ana M. Herrera, Antonio Ramos, Godofredo García, Gustavo A. Hirata, José A. Luna, Roberto C. Carrillo, Jorge A. Rodríguez, Mario Robles, Yani D. Ramírez and Guillermo Martínez. We have implemented several changes in the text to answer all indications requested by the assistant editor and reviewer. We believe that their observations allowed us to improve substantially the quality of our work, and hence we manifest our gratitude to the professional job they did in the review of our manuscript.
In the revised manuscript, the yellow underlined sentences correspond to the corrections of the second reviewer; in addition to this, a thorough review of the manuscript was carried out taking into account the comments of the second reviewer. The manuscript was also proofread by a native grammar checker.
Yours Sincerely
Dr. Erick Gastellóu
erick_gastellou@utpuebla.edu.mx
Corresponding author
Comments of the second reviewer:
The authors claim that “the photoluminescence spectrum for Mg-Zn co-doped GaN powders showed two fundamental emissions, the first located at 3.40 eV (364.70 nm) related to band-to-band transition, and a second emission located in a range from 2.80 eV to 2.88 eV (442.85 nm – 430.55 nm), which is characteristic of Mg-doped GaN and Zn-doped GaN powders”.
It is known that Mg introduces two acceptor levels in GaN. The acceptor bound excitons give two PL peaks at 3.466 eV (unstable in p-GaN) and 3.454 eV. The most common radiation bands in Mg-doped GaN are the BL bands with the peak of about 2.9 eV, and the UVL band, whose peak is about 3.2 eV. Sometimes, the GL band of about 2.4-2.5 eV can appear. It seems that the authors do not identify all the bands related with Mg impurity in GaN. Moreover, it is not entirely clear whether the authors distinguish the difference between the bands associated with Mg and Zn.
Answer: The c peak of Figure 7 was decomposed to observe the emissions of 2.90 eV corresponding to GaN doped with Mg, as well as the emission of 2.80 corresponding to Zn-doped GaN (Figure 8). Therefore, the emissions corresponding to Mg and Zn are identified, as suggested by reviewer number 2.

Reviewer 3 Report
In this work, Mg-Zn codoped GaN powders via nitridation of a Ga-Mg-Zn metallic solution at 1000 °C for two hours in NH3 flow were obtained. X-ray diffraction patterns for the Mg-Zn codoped GaN powders showed a crystal size average of 46.88 nm. Scanning electron microscopy micrographs showed an irregular shape with a structure like ribbons and an average length of 8.63 µm. Energy dispersive spectroscopy and X-ray photoelectron spectroscopy measurements demonstrated the elemental contributions of magnesium and zinc as codopant atoms into GaN powders. Transmission electron microscopy characterization showed a greater scattering of impurity atoms for the Mg-Zn codoped GaN powders compared to undoped GaN powders. Photoluminescence spectrum showed a fundamental emission located at 3.40 eV( 364.70 nm), which was related to band-to-band transition, besides a second emission located in a range from 2.80 eV to 2.88 eV (442.85 nm – 430.55 nm), was related as a characteristic of Mg-doped GaN and Zn-doped GaN powders. Finally, Raman scattering demonstrated a shoulder at 648.05 cm-1 , which could indicate the incorporation of the Mg and Zn codopants atoms into the GaN structure.There is a certain amount of creativity in this work. For this work to improve the following problems can be accepted later.
(1) The authors are suggested to compare and discussion some recent works, (i.e. Ultrafast Science, 2022, 9767251, 16, 2022; Ultrafast Science, 2022, 9870325, 6, 2022; Ultrafast Science, 3, 0006, 2023; Laser Photon. Rev. 13, 1800333, 2019; Phys. Rev. Lett., 123, 093901, 2019).
(2) The reference logo can be typeset at the top right of the last letter at the end of the sentence.
(2) Figure 1 can be placed above heading 2.2 to make the article easier to read.
(3) In the introduction, “Codoping is an important strategy used for tuning the dopant in electronic, and magnetic properties, besides of enhances the dopant solubility, increasing the activation by lowering the ionization energy of acceptors and donor, in this way increasing the carrier’s mobility” Some related work needs to be discussed in the manuscript. (Applied Materials & Today, 2022,28,101546)
(4)The relationship between the surface morphology of banded structures in FIG. 3b and the combination of Mg and Zn into GaN by diffusion can be further studied.
Author Response
March 28th, 2023
Reviewer 3
Manuscript ID: materials-2270136
Dear Reviewer 3
Enclosed you will find the revised version of the manuscript: “Obtaining of Mg-Zn codoped GaN powders via nitridation of the Ga-Mg-Zn metallic solution and their structural, and optical properties", by Erick Gastellóu, Rafael García, Ana M. Herrera, Antonio Ramos, Godofredo García, Gustavo A. Hirata, José A. Luna, Roberto C. Carrillo, Jorge A. Rodríguez, Mario Robles, Yani D. Ramírez and Guillermo Martínez. We have implemented several changes in the text to answer all indications requested by the assistant editor and reviewer. We believe that their observations allowed us to improve substantially the quality of our work, and hence we manifest our gratitude to the professional job they did in the review of our manuscript.
In the revised manuscript, the yellow underlined sentences correspond to the corrections of the third reviewer; in addition to this, a thorough review of the manuscript was carried out taking into account the comments of the third reviewer. The manuscript was also proofread by a native grammar checker.
Yours Sincerely
Dr. Erick Gastellóu
erick_gastellou@utpuebla.edu.mx
Corresponding author
Comments of the second reviewer:
(1) The authors are suggested to compare and discussion some recent works, (i.e. Ultrafast Science, 2022, 9767251, 16, 2022; Ultrafast Science, 2022, 9870325, 6, 2022; Ultrafast Science, 3, 0006, 2023; Laser Photon. Rev. 13, 1800333, 2019; Phys. Rev. Lett., 123, 093901, 2019).
Answer: All the references indicated by the reviewer have been included in the manuscript, for which we would like to thank reviewer number 3 for his suggestions.
(2) The reference logo can be typeset at the top right of the last letter at the end of the sentence.
We sincerely apologize to reviewer number 3, but the way to indicate the reference is part of the word template (format) of the materials journal, so we cannot modify the presentation of the references.
(2) Figure 1 can be placed above heading 2.2 to make the article easier to read.
Answer: La etiqueta del subtitulo "2.1 Mg-Zn codoped GaN powders" fue colocada en el encabezado de la Figura 1, tal como lo sugierió el revisor número 3 para identificar la figura con los parrafos correspondientes.
(3) In the introduction, “Codoping is an important strategy used for tuning the dopant in electronic, and magnetic properties, besides of enhances the dopant solubility, increasing the activation by lowering the ionization energy of acceptors and donor, in this way increasing the carrier’s mobility” Some related work needs to be discussed in the manuscript. (Applied Materials & Today, 2022,28,101546)
Answer: The reference Applied Materials & Today, 2022,28,101546 was added to the paragraph indicated by reviewer 3, so we appreciate your suggestion.
(4) The relationship between the surface morphology of banded structures in FIG. 3b and the combination of Mg and Zn into GaN by diffusion can be further studied.
Answer: As indicated by reviewer number 3, a proposal to obtain the structures like ribbons that form the Mg-Zn codoped GaN powders was added to the SEM analysis.

Reviewer 4 Report
The comments of the manuscript are shown in a separate file.

Author Response
March 28th, 2023
Reviewer 4
Manuscript ID: materials-2270136
Dear Reviewer 4
Enclosed you will find the revised version of the manuscript: “Obtaining of Mg-Zn codoped GaN powders via nitridation of the Ga-Mg-Zn metallic solution and their structural, and optical properties", by Erick Gastellóu, Rafael García, Ana M. Herrera, Antonio Ramos, Godofredo García, Gustavo A. Hirata, José A. Luna, Roberto C. Carrillo, Jorge A. Rodríguez, Mario Robles, Yani D. Ramírez and Guillermo Martínez. We have implemented several changes in the text to answer all indications requested by the assistant editor and reviewer. We believe that their observations allowed us to improve substantially the quality of our work, and hence we manifest our gratitude to the professional job they did in the review of our manuscript.
In the revised manuscript, the yellow underlined sentences correspond to the corrections of the fourth reviewer; in addition to this, a thorough review of the manuscript was carried out taking into account the comments of the fourth reviewer. The manuscript was also proofread by a native grammar checker.
Yours Sincerely
Dr. Erick Gastellóu
erick_gastellou@utpuebla.edu.mx
Corresponding author
Comments of the fourth reviewer:
The authors reported an Mg-Zn codoped GaN powder by using nitridation of the Ga-Mg-Zn metallic solution. They studied the characteristics of materials and optoelectronics of Mg-Zn codoped GaN powder. Although authors successfully codoped Mg and Zn in GaN powder, there are comments to the report.
- The authors might measure the concentration or composition of Mg and Zn in the GaN powder.
Answer: EDS studies showed the atomic percentages of zinc to be 0.60%, and magnesium to be 0.32%m, as shown in the electron microscopy analysis, section 3.2.
- What are the electrical properties of the Mg-Zn codoped GaN powder?
Answer: For the moment, only the structural and optical properties have been studied. However, once the targets are manufactured using the powders obtained in this work, thin films of Mg-Zn codoped GaN will be deposited, in which electrical measurements will be made to determine hole and electron concentrations.
- Why did the Cu appear in both undoped and Mg-Zn codoped GaN powder?
Answer: The copper contributions that were presented in the EDS analysis belong to the electron microscope sample holder, because the powder samples were small. The contribution of copper is mentioned in section 3.2 electron microscopy.
- The authors need to explain why the Mg-Zn codoped GaN powder showed the band edge transition and the undoped GaN powder showed the transition of excitons bound to structural defects.
Answer: Reviewer number four's comment was included in the rewrite of the photoluminescence analysis, where the emission of the dopants was even deconvoluted to find out which emission belongs to Mg and which belongs to Zn.

Round 2
Reviewer 4 Report
The authors revised their manuscript with the reviewers' comments. The manuscript is now clear to readers and good for publication.